

# Changing patterns of extreme water levels in urbanizing plain river network region of Taihu Basin, China: characteristics and causes

Yuefeng Wang, Youpeng Xu, Yu Xu, Song Song, Guang Li, Lei Wu

School of Geographic and Oceanographic Sciences, Nanjing University, Nanjing, China

*Correspondence to*: Youpeng Xu (xypnju@163.com)

**Abstract**. Water level is an indicating factor in flood control in the plain river network region of Taihu Basin (PRNRTB). It is mainly influenced by climate change and human activity. In this study, the annual and seasonal variations of extreme water level from 1960 to 2012 were analyzed based on daily water level of eight stations in the PRNRTB. The modified Mann-Kendall test and sequential cluster analysis are used to detect trends and points of

abrupt change. The results indicated that the extreme water level shows a significant increase at a regional scale. The increases in extreme high water level (EHWL) and extreme low water level (ELWL) were 0.007 and 0.01 m per year, respectively. Detected points of abrupt change was around 1988 for the region and most stations, which correspond to the period of intensive human activities in this region. The changes in average annual EHWL and ELWL between 1989 and 2012 are, respectively, 7.8% and 12.7% higher than that between 1960 and 1988. Meanwhile, contributions of

precipitation and human activity were also assessed in three individual periods (1989–2012, 1989–2000, and 2000–2012). Between 1989 and 2012, the contribution from human activity increased from 20.5% to 70.3% for EHWL, while human activity was always the main driver responsible for the increase in ELWL in that period. In addition, a thorough discussion is included about the potential driving force on the extreme water level in the PRNRTB. Human activities are suggested to have played more and more important roles in the extreme water level changes since the late

1980s. The results of the study would provide support in water resources management and floods control in urban development.

## 1 Introduction

Urbanization is a global phenomenon that poses profound threats to the local environment. Over the past few decades, a wide range of environmental damages have been linked to the urbanization: impervious surfaces increasing,

hydrological alterations, water infrastructure construction and water environment degradation (Allaire et al., 2015; Huong and Pathirana, 2013). Such modifications to the landscape will result in changes to the hydrologic cycle and watershed processes (Hao et al., 2015). Besides, the occurrence of frequent hydrological extremes, such as drought and flood hazards, will threaten agricultural production and socio-economic development (Milly et al., 2002; Thibault and Brown, 2008). Water levels, like other hydrological variables in a river basin, are one of the most important indicators

to evaluate changes of water resources and flood disasters (James, 2016). In recent years, the variations in extreme





water levels (i.e. maximum and minimum water levels) are of increasing global attention because of the increased risk of floods and droughts on local or regional scales, as well as an increasing or decreasing water resources at the continental scale that have been observed (Zhang et al., 2009; Doell and Zhang, 2010; Assani et al., 2014). Meanwhile, water levels also have an important impact on the critical functions of river basins, such as ecosystem stability, water

environment and engineering construction (Wen et al., 2011; Reid et al., 2013).

Climate variations and human activities are commonly recognized as two major factors impacting the long-term hydrological alterations (Yao et al., 2014; Wang et al., 2015). A large of studies on variation in water levels of watersheds in the world and their relations with the climate variability and human activities have been carried out (Ou et al., 2011; Li and Zhang, 2014; Cochrane et al., 2014). However, more and more researchers believe that both

climatic and anthropogenic factors should be responsible for the variation of water levels in recent decades (Scuderi et al., 2010; Yin et al., 2013; Yuan et al., 2015). Climatic variation is one of important driven factors affecting water levels through changes in precipitation, temperature and evaporation (Song et al., 2014). Anthropogenic influences, such as dam construction, reservoir operation, destruction of vegetation and river network reduction would alter hydrological process and water levels (Gibson et al., 2006; Yuan et al., 2015). Zhang et al. (2006) investigated the

annual maximum runoff and water level of the middle Yangtze River, both of which showed a significant upward trend. Yin et al. (2009) analyzed the maximum water levels changes of Taihu Basin from 1954 to 1999, coming to the conclusion that the rise of maximum water levels mainly determined by human activities since 1980s. Chen et al. (2009) analyzed the change points of extreme water level and possible causes in the Pearl River Delta. According to Xu et al. (2014), temporal and spatial variation of extreme water level in Hang-Jia-Hu plain is mainly caused by rapid

urbanization. What's more, some other researches related to variation in water levels are mainly focusing on lake basins in China, such as Dongting Lake (Yuan et al., 2015), Poyang Lake (Li et al., 2014) and Hongze Lake (Yin et al., 2013). However, very few relevant studies have been performed in plain river network region.

The criss-cross river network with a density of 3.2 km/km$^2$ in the plain river network region of Taihu Basin (PRNRTB) is a region with one of the most complicated drainage systems in the world (Deng et al., 2016). Compared

to mountainous basins, there are variable flow directions and cross-river networks in the PRNRTB. Because of the low and flat topography and gentle slope, water level becomes a key factor to investigate the long-term hydrological variation under climate changes and human activities in this region (Xu et al., 2014). Due to uneven spatial-temporal distribution of precipitation, frequent floods and droughts in this region have threatened its economic development. For example, the largest flood disaster occurred in 1999, which resulted in damages with direct economic loss of $16

billion damages (Wang et al., 2010). The PRNRTB is also one of the major economic centers of China with high-density population. Encompassing Suzhou, Wuxi and Changzhou city, this area is undergoing the most rapid urbanization development, which has caused significant changes in local environment, such as flood disaster, river reduction and water quality deterioration. Some relevant studies about environment changes and hydrological variations have been carried out in this region (Deng et al., 2015; Wang et al., 2016). However, there are few

comprehensive researches on long-term changes in extreme water level and the relevant driving forces analysis in the





PRNRTB. Therefore, the objectives of the study are designed to (1) reveal the annual and seasonal variation of extreme water level during the period of 1960-2012; (2) detect change points of different extreme water level; and (3) determine whether climate or anthropogenic factor should be responsible for the variability of extreme water level in PRNRTB. Such a study will provide the theoretical basis for the local water resource management and floods control.

## 2 Study area and data source

### 2.1 Study area

This study is conducted in the PRNRTB, which is located in the center of the Yangtze River Delta and covers a total land area of 7,929 km$^2$, 2-4 m above sea level (Fig. 1). It mainly includes Wuchengxiyu and Yangchengdianmao water conservancy region. As one of the famous water-towns in the world, this PRNRTB is characterized by the presence of many rivers and lakes, which become the linkage of the Yangtze River and Tai Lake. This region is controlled by the East Asia Monsoon with four distinct seasons. The average annual temperature is 16 ℃, and the average annual rainfall is 1,180 mm. The precipitation is mainly concentrated during the flood season, including plum rain (May to June) and typhoon rain (July to September), which accounts for 60 % of the annual precipitation (Yang et al., 2014). As Fig. 2 shown, the changes of average water level are similar to precipitation. The high value of water level mainly occurs from June to October and the maximum value appears in July.

Rapid urbanization and economic growth have taken place since 1980s. Su-Xi-Chang (i.e. Suzhou, Wuxi and Changzhou city) economic zone has become one of the important economic centers in China. Currently, the region contributes 7.0 % GDP of China although it is only 0.1% of the total land area of China (Ning et al., 2010). It is noteworthy that the increasing population, rapid industrialization and highly urbanization have made a considerable pressure on the PRNRTB. The proportion of farmland areas decreased from 44.94% to 37.62% while industrial and urban areas increased by 4.67% annually in the same period (Wang et al., 2010). River systems also have been severely disturbed with the process of urbanization (Deng et al., 2016). More and more water engineering facilities, including sluices, pumps and dikes, have been constructed for flood defense (Liu et al., 2013). Therefore, further studies about variations of extreme water level under dramatic changes of the underlying surface are still necessary in the PRNRTB.

### 2.2 Data Sources

Water level is an important basis for flood control in the PRNRTB. The daily water level dataset used in this paper covered the period from 1960 to 2012 and were collected from 8 water level stations within the PRNRTB (Table 1). All stations are located in regional backbone river and the mean water level is about 2.60-3.30 meters. Annual daily extreme water levels, including extreme maximum high water level (EHWL) and extreme maximum low water level (ELWL) were collected. The thiessen polygons method (Jones and Hulme, 1996) was applied to calculate regional extreme water level series, which represent the temporal evolution of the water level in the entire basin. Additionally, precipitation data of 24 stations were used to calculate the regional precipitation of the PRNRTB during 1960-2012 (Fig. 1). All these data were obtained from Taihu Basin Management Bureau of China and have been used in some





studies (Yin et al., 2012; Yang et al., 2014). Moreover, some other data including land use, river network and dikes of the PRNRTB are also collected to analyze the effects of human activities on extreme water level.

## 3 Methodology

### 3.1 Analysis of extreme water level trends and change points

**3.1.1 Trend analysis**

The nonparametric Mann-Kendall (M-K) test is one of the most widely used methods for hydro-meteorological time series, and is particularly applicable in detecting trends of precipitation, temperature, and water level (Mann 1945; Kendall 1975). Based on the M-K test method, the $\beta$ slope of the series can be computed by Thiel-Sen method.

$$\beta = Median(\frac{x_j - x_i}{j - i}) \quad \forall \, 1 < i < j < n$$

Where, $x_j$ and $x_i$ are the observed values in the $j$-th and $i$-th year ($j > i$), respectively. However, certain hydro-meteorological time series may frequently display statistically significant serial correlation. This may lead to a disproportionate rejection of the null hypothesis of no trend whereas the null hypothesis is actually true (Storch and Navarra, 1999). To eliminate the influence of serial correlation on M-K results, trend-free pre-whitening (TFPW) was used for the time series before M-K analysis, which was proposed by Yue et al. (2002). The new time series was

obtained as follows:

$$x'_i = x_i - (\beta \times i)$$

The lag-1 series correlation ($r_1$) value of this new time data set was calculated and used to determine the residual series as:

$$y'_i = x'_{i+1} - r_1 \times x'_i$$

The value of $\beta \times i$ was added again to the residual data set as follows:

$$y_i = y'_i + (\beta \times i)$$

The $y_i$ series was subjected to trend analysis. In this study, the 0.05 and 0.01 significant level was used as the threshold to classify the significance of positive and negative M-K trends.

**3.1.2 Change point analysis**

The intensifying human activities in land use and river networks disturbed the natural hydrological cycle in the PRNRTB, which could have an impact on the characteristic of the water level. The sequential cluster analysis is an effective method to extract hydrologic series change points (Yue et al., 2014). Once the change point is detected through the test, then the water level series would be divided into two homogeneous groups, which represent





heterogeneous characteristics from each other.

For a water level series $x_1, x_2, \ldots, x_n$, the sequential cluster analysis is shown as follows:

$$V_\tau = \sum (\alpha_t - \overline{\alpha_\tau})^2$$

$$V_{n-\tau} = \sum (\alpha_t - \overline{\alpha_{n-\tau}})^2$$

$$S_n(\tau) = V_\tau + V_{n-\tau}$$

$$S_n^* = \min[S_n(\tau)]$$

Where, $\alpha_t$ is the hydrological variable (i.e., water level); $\tau$ is the change point; $\overline{\alpha_\tau}$ means the hydrological series

mean value before change point; $\overline{\alpha_{n-\tau}}$ means the hydrological series mean value after change point; $V_\tau$ and $V_{n-\tau}$ are the

sum of the squared deviations of hydrological series before and after the change point year $\tau$, respectively; and $S_n^*$ is

the smallest sum of squared deviations where the optimal change point appears.

### 3.2 Estimating method of water level variability attribution

Changes in water level are mainly affected by precipitation and human activity in the PRNRTB. The intensive human activities took place in this region after 1980s. Although they are inter-related, we attempt to assess their individual impact on water level. In this study, a method is used to identify the contributions of precipitation and human activities to changes in water level (Wang et al., 2008). A change in water level can be calculated as follows:

$$\Delta H_T = H_A - H_O$$

$$\Delta H_H = H_A - H_{HP}$$

$$\Delta H_P = H_{HP} - H_O$$

Where, $\Delta H_T$ indicates the observed change in mean annual water level between two different periods; $H_O$ is the average annual water level during baseline periods; and $H_A$ is the average annual water level change during the activity period. $H_{HP}$ is the natural water level in the activity period; $\Delta H_H$ and $\Delta H_p$ indicate the changes in the mean annual water level due to human activity and climate variation, respectively. We have an assumption that there is no significant human activities occurred in the baseline period. First, an equation is established between annual precipitation and water level in the baseline period. And then, this equation is used to estimate the value of $H_{HP}$ in the activity period. The relative contribution of precipitation variation and human activities on water level can be expressed in percentage



as follows:

$$\eta_P = \frac{\Delta H_P}{\Delta H_T} \times 100\%$$

$$\eta_H = \frac{\Delta H_H}{\Delta H_T} \times 100\%$$

Where, $\eta_P$ and $\eta_H$ are the percentage of the precipitation variation and human activity impact on the water level, respectively.

## 4 Results and discussions

### 4.1 Analysis of the variability of extreme water level

### 4.1.1 Inter-annual variation of extreme water level

The inter-annual variations of extreme water level in the PRNRTB during the period of 1960-2012 were analyzed by linear trend and M-K test. As showed in Fig. 3, there are strong upward trends for the both water level series. For EHWL, the maximum value was 4.55 m (occurring in 1962), and the minimum value was only 3.10 m (occurring in 1978). The slope of linear regression is 0.007 ($R^2 = 0.13$, $p < 0.05$). It can be seen that the EHWL was less than 4.0 m before 1989 (except 1962) and its frequency above 4.0 m during the 1990s was higher than other decade. Due to the continuous rainstorms in the 1990s, the flood disasters occurred frequently, including the largest flood in 1999. The ELWL had also an increasing trend with slope of 0.01 ($R^2 = 0.68$, $p < 0.05$). The maximum and minimum of the ELWL occurred in 2012 and 1978, respectively. From an ecological standpoint, the rising ELWL could dilute the pollutants and improve the water quality of the PRNRTB. However, it also could contribute to floods generation to some extent.

Considering the effect of serial correlations on the trend test results, the TFPW procedure was used for the time series before M-K analysis. The M-K test results of extreme water level for the PRNRTB are plotted in Fig. 4. We can understand the variation trend according to the curve of statistic UF. The positive UF indicates that the water level has an increasing trend, and the negative one indicates the opposite trend. As shown in Fig. 4a, the EHWL showed an increasing trend before 1963, afterwards, a declining trend from 1964 to 1989, followed by a growing trend which passed the 0.05 significance level after 2000. The changing curves of statistic UF for ELWL consisted of two stages: fluctuation around zero before 1980, and then a rapid upward trend until 2012. The increasing trend passed the 0.05 significant levels since the late 1980s (Fig. 4b). Generally, significant changes of the extreme water level in the PRNRTB occurred after the late 1980s, which is related to climate change and intensive human activity in this region, especially the rapid urbanization since the 1980s.

As shown in Table 2, the serial correlations of water level series were also detected at individual station. The lag-1 $R$ of the extreme water level for most stations is positive. It is obvious that lag-1 $R$ of ELWL is higher than that of EHWL.





The variation trend and slope of extreme water level for each station are also listed in Table 2. It is shown that both EHWL and ELWL at all stations have increasing trends. The trends of ELWL at each station are more than 0.01 significance levels while only five stations are beyond 0.01 significance levels for the trends of EHWL. Spatially, the increasing trends of the stations in Yangchengdianmao region are slightly higher than these in Wuchengxiyu region. It is mainly because there was much more human activity in Yangchengdianmao region due to its economic location.

### 4.1.2 Change point in extreme water level

The sequential cluster analysis is applied to detect the abrupt changes of average extreme water level. It can be clearly seen that the abrupt change for the EHWL and ELWL of the whole PRNRTB region during 1960-2012 occurred in 1988 (Fig. 5). As shown in Fig. 5a, the average EHWL increased from 3.72 m during the period of 1960-1988 to 4.01 m during the period of 1989-2010 with an increase of 7.80 %. The abrupt increment was mainly because of intensive human activities, especially the increasing impervious area and declining river network after the 1980s, which would contribute to the runoff coefficient (Xu et al., 2014). Meanwhile, the change point of the ELWL also appeared in 1988 (Fig. 5b). The average ELWL rose from 2.51 m over the period 1960-1988 to 2.83 m in 1986-2010 with an increase of 12.75 %. As Xu et al. (2010) reported, a large number of water conservancy projects have been built for flood control in this region since the 1990s. However, in order to improve the urban water environment, a higher normal water level often is maintained by local government departments, which is an important reason for the increasing of ELWL in the PRNRTB. Furthermore, the change points of each station were also investigated (Table 2). It can be seen that the timing of abrupt change occurred in 1987-1990 (around 1988) for most stations. Therefore, considering the variation period of extreme water level in the PRNRTB, we divided the time series (1960-2012) into two distinguishable periods, namely 1960-1988 and 1989-2012.

### 4.1.3 Seasonal variation of extreme water level

To further explore the variation in extreme water level, monthly EHWL and ELWL in the PRNRTB region were also analyzed. As shown in Table 3, the trends of all water level series are more than 0.01 significance levels except EHWL in October. The maximum trend of EHWL and ELWL occurred in March and February, respectively (Table 3). What's more, it should be notable that the upward trends of EHWL in rainy season (May to October) are lower than these in dry season (November to April). As we all know, the EHWL usually accompanies by rainfall, which is mainly determined by precipitation process and shows certain randomness. The trends of monthly ELWL don't show significant differences with each other. This is because ELWL usually occurs in normal water level which is mainly controlled through operations of water conservancy facilities.

The monthly variations of extreme water level during the three periods (1960-1988, 1989-2012 and 1960-2012) are shown in Fig. 6. The monthly extreme water level exhibits a seasonal pattern in the PRNRTB. The peak of extreme water level occurs in July, and the lowest value occurs during the winter months. It was evident that the evolutions of monthly extreme water level were approximately similar among the three periods. However, there was significant difference for monthly extreme water level during the period of 1960-1988 (baseline period) and the period of





1989-2012 (activity period). The monthly water level during 1989-2012 had an overall upward shift than before. The monthly EHWL and ELWL during the period of 1989-2012 were respectively 6.06-15.28 % and 6.44-15.22 % higher than that during the period of 1960-1988. The increment of extreme water level in dry season was larger than that in rainy season (Fig. 6a and b). In comparison, the monthly average precipitation did not show the similar variation like the extreme water level during the different periods (Fig. 6c). Therefore, it can be inferred that the alteration of monthly extreme water level might be influenced by a combination of human activity and precipitation.

## 4.2 Quantifying the precipitation and human activity effects on extreme water level

The variation in water level is a result of catchment processes, which is affected by many factors including precipitation and human activity (Yuan et al., 2015). Some studies indicated that intensive human activities, such as land use change, river network reduction and water conservancy construction had occurred in the PRNRTB since the late 1980s (Yin et al., 2012; Huang et al., 2015). To quantify the influences of precipitation and human activity on extreme water level, the period from 1960 to 1988 is regarded as baseline period because previous studies indicated that the PRNRTB was not significantly affected by human activity prior to the 1980s (Wang et al., 2010). Therefore, human activity during the baseline period is assumed to be negligible. To further examine the contribution to extreme water level during different periods, the activity period was divided into two shorter periods (i.e. 1989-2000 and 2001-2012). The linear regression method was applied to construct the equations between extreme water level and precipitation (Fig. 7). The correlation coefficient between EHWL and precipitation during 1960-1988 was 0.80, which is the higher than that during 1989-2000 (0.79) and 2000-2012 (0.27). The correlation coefficients between ELWL and precipitation were 0.43, 0.13 and 0.11 in the three periods respectively. The results indicate that the response relationship between extreme water level and precipitation gradually weakened due to some other factors.

The equations of baseline period (1960-1988) are used to estimate the natural extreme water level during activity periods. Table 4 summarizes the contributions of precipitation and human activity to changes in extreme water level during different periods. Generally, both precipitation and human activity had positive effects on EHWL and ELWL compared to the baseline period. Precipitation variation played a dominant role in the increasing of EHWL in 1988-2000 and 1989-2012, the contributions of which were 79.5 % and 53.3 % respectively. The contribution of precipitation dropped to 29.7% and the human activity played a more important role for the increasing of EHWL during 2001-2012. Clearly, the effect of human activity was the dominant factor for the increase of ELWL in all the periods. The contribution from human activity became larger from 54.8 % in 1989-2000 to 80.1 % in 2001-2012. It should be noticed that the increasing extents of extreme water level caused by human activity become larger and larger in the PRNRTB.

## 4.3 Discussions

### 4.3.1 Impacts of precipitation on the alterations of extreme water level

Water level is an important parameter and impacts critical functions of the PRNRTB, such as construction of levees, flood control and water environment (Lin, 2002). Precipitation, as one of important climate factors, has a direct impact





on the variation of extreme water level in this region, especially for EHWL, which is usually accompanied by heavy rainfall. Variation of annual precipitation in the PRNRTB during the period of 1960-2012 was shown in Fig. 8. The annual precipitation had an upward trend with a slope of 2.06 mm per year. The annual precipitation fluctuated during the period of 1960-1988 with an average precipitation of 1015 mm while it increased during the period of 1989-2012

with average precipitation of 1095 mm. However, there was no obvious change point for annual precipitation like extreme water level. The statistic $Z$ of annual precipitation of the PRNRTB was 1.20, which was not significant and lower than that of extreme water level. Moreover, the correlation coefficient between extreme water level and annual precipitation began to decrease during activity period. A conclusion can be made that precipitation might be a main factor responsible for variation of water level before 1988 while human activity should be considered as an important

factor after 1988.

On the other hand, changes of precipitation structure and extreme precipitation are also important factors to impact extreme water level. Deng et al. (2014) analyze the heterogeneity of precipitation at multi-time scale in the Taihu Basin, which indicated that heterogeneity and concentration of precipitation in wet season will continue the rise trend. According to (Han et al., 2015), rainy day with short duration (2-4 days) show upward occurrence and fractional

contribution to annual precipitation and are characterized by intensified precipitation, which may contribute to the increasing of EHWL. Since the 1990s, the water level has surpassed the warning level more frequently. What's more, due to the occurrence of extreme climate events, the frequency of flood disasters in Taihu Basin has been increasing in recent years, resulting in more drowned areas and economic loss (Yu et al., 2013). Therefore, much more attention should also be paid to the variation of extreme precipitation by relevant government departments.

**4.3.2 Impacts of human activity on the alterations of extreme water level**

Human activities, such as landscape changes, hydraulic projects construction, river channelization, may impact on the alteration of water level (Assani et al., 2014; Yuan et al., 2015). Since the late 1980s, more and more human activities have occurred in the Taihu Basin (Xu et al., 2010; Ning et al., 2010). Due to the rapid urbanization in the PRNRTB, land use changed dramatically and impervious surfaces increased significantly. The general trend of land use in the

PRNRTB is characterized by reduced paddy land and increased urbanized land from 1991 to 2009 (Fig. 9). The proportion of paddy land decreased from 74.28 % in 1991 to 41.54 % in 2009 with an annual declining rate of 1.17 %. Total urbanized land increased by 240km$^2$, with an annual growth rate of 85.9 km$^2$ per year during the same period. Land use changes have many adverse influences on the functions of the PRNRTB, such as surface runoff increase and water quality deterioration (Huang et al., 2015). However, as cities are dynamically expanding, the continuous increase

of impervious surfaces will result in higher water level in flash floods and increase the probability of disasters.

River networks also play an important role on water level and flood regulation in the Taihu Basin. Wang et al. (1999) reported that storage capacity of river networks accounted for about 50% of the entire basin, which could significantly mitigate flood during high water level. However, river networks have been damaged during the rapid urbanization in recent decades, which would alter the regional hydrological process to some extent (Song et al., 2016). Table 5 shows

four characteristic indicators of river networks in the PRNRTB over the past 50 years. There was an obvious reduction



in river length and water area from the 1960s to 2010s. The drainage density and water surface ratio decreased by 12.57% and 20.21% during the past 50 years respectively, the reduction of which from the 1980s to 2010s was higher than that during the 1960s to 1980s. Deng et al. (2016) analyzed the evolution of the distribution pattern of river system and the results reflected that the low-order river experienced a dramatic decrease in the PRNRTN. Generally, the disappearance of rivers may lead to the excess runoff exceeding the present channel capacity and result in local flash floods (Yang et al., 2016). These changes of river networks also were an important factor for the increasing of extreme water level in this region.

In recent decades, for the sake of flood control, a large number of hydraulic projects (sluices and pumps) have been constructed above the rivers, which resulted in emergence of polder districts in the PRNRTB (Xu and Yang, 2013). The sluices and pumps can exert great impacts on water level and hydrologic connectivity (Cochrane et al., 2014). According to the statistical data, a total of 2488 conservancy facilities have been constructed during the past four decades, most of which were built in the 1990s (706) and 2000s (1484). Since the floods of 1999, the local governments began to strengthen the construction of polder districts, which increased the drainage capacity of polder districts. As a result, a lot of runoff was discharged from polder district through pumps during the heavy rainfall, which would make water level of outer river sharply increase and transfer the flood risk to the entire basin. On the other hand, water level in polder districts depends largely on the opening and closing operations of the many sluices and pumps in dry season. Thus, these hydraulic projects also played an important role in improving water environment of polder districts through artificial regulation. Although a higher normal water level would increase the ability to dilute pollutants to some extent, yet it also could make the flood risk become larger. Therefore, some measures must be taken to pre-fall water level of polder districts before a heavy rainstorm coming.

## 5 Conclusions

In this study, annual and monthly variations of extreme water level (EHEL and ELWL) in the PRNRTB during the period of 1960-2012 were analyzed to evaluate the potential impacts of climate change (primarily through precipitation) and human activities under rapid urbanization.

The results showed that extreme water level displayed a significant increasing trend at regional scale and the trend of ELWL is higher than that of EHWL. The average annual increasing rate of EHWL and ELWL were 0.007 and 0.01 m per year, respectively. The same characteristics were also detected from each individual station.

The results of the sequential cluster analysis indicated that the abrupt change mainly occurred in the late 1980s (in about 1988), which also corresponds to the period of most intensive human activities in this region. The amounts of change in average annual EHWL and ELWL of 1989-2012 (activity period) are 7.8 % and 12.7 % respectively, higher than that of 1960-1988 (baseline period). The effects of precipitation variation and human activity on extreme water level varied temporally. The contribution from human activity increased from 20.5% (1989-2000) to 70.3% (2001-2012) for EHWL while the human activity was always the main driver responsible for the increasing of ELWL since 1989.



Understanding the variation regularity of water level and the potential drivers can provide insights into regional flood control and water environment protection. A thorough discussion was made about the potential driving force from precipitation and human activity on extreme water level in the PRNRTB. Although the positive trend of annual precipitation is not significant and lower than that of extreme water level, it is always an important factor for extreme

water level changes. Nevertheless, due to rapid urbanization, a variety of human activities have effects on the increasing of extreme water level in this region. It should be noticed that the role of human activity in water level change and flood hazards in recent decades. The continuously increasing extreme water level in the PRNRTB would increase flood risk to the local cities. Therefore, some other measures should be taken to coordinate the relationship between cities floods and basin floods through hydraulic projects operations when encountering a flood in the

PRNRTB.

*Author contributions*: The authors collectively designed the experiments and interpreted the results. Wang analyzed the data and wrote the paper, that all authors commented on.

*Acknowledgements*:This research was financially supported by the National Natural Science Foundation of China (Grant No. 41371046), the Water Conservancy Science and Technology Foundation of Jiangsu Province (Grant No. 2015003), the Commonweal Specialized Programs for Scientific Research, Ministry of Water Resources of China (Grant No. 201201072), and the key program of Jiangsu Natural Science Foundation (Grant No. BK20131276, BK20150584).



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





Table 1 Detail information of the water level stations in the PRNRTB

| Station name | Longitude (°E) | Latitude (°N) | Mean water level (m) | Series length | River name |
|---|---|---|---|---|---|
| 1. Wuxi | 120.31 | 31.57 | 3.07 | 1960-2012 | Jinghang Channel |
| 2. Chenshu | 120.54 | 31.73 | 3.10 | 1960-2012 | Zhangjiagang River |
| 3. Qingyang | 120.25 | 31.75 | 3.20 | 1960-2012 | Xicheng Channel |
| 4. Changzhou | 119.98 | 31.77 | 3.33 | 1960-2012 | Jinghang Channel |
| 5. Suzhou | 120.64 | 31.29 | 2.80 | 1961-2012 | Jinghang Channel |
| 6. Kunshan | 120.97 | 31.39 | 2.66 | 1971-2012 | Loujiang River |
| 7. Pingwang | 120.63 | 31.00 | 2.86 | 1960-2012 | Taipuhe River |
| 8. Changshu | 120.75 | 31.64 | 2.92 | 1960-2012 | Baimaotang River |



Table 2 M-K test results and change point for the extreme water level of each station in the PRNRTB

| No. | Station name | Parameter | Lag-1 $R$ | $\beta$ | M-K test | | Change point |
|-----|-------------|-----------|-----------|---------|-----|-------------|--------------|
| | | | | | $Z$ | Significant | |
| 1 | Wuxi | EHWL | 0.14 | 0.0129 | 3.64 | ** | 1988 |
| | | ELWL | 0.33 | 0.0113 | 6.51 | ** | 1989 |
| 2 | Chenshu | EHWL | 0.05 | 0.0071 | 2.28 | * | 1990 |
| | | ELWL | 0.30 | 0.0120 | 6.84 | ** | 1993 |
| 3 | Qingyang | EHWL | 0.09 | 0.0086 | 2.36 | * | 1989 |
| | | ELWL | 0.43 | 0.0111 | 5.71 | ** | 1989 |
| 4 | Changzhou | EHWL | -0.13 | 0.0056 | 1.76 | | 1986 |
| | | ELWL | 0.16 | 0.0107 | 6.67 | ** | 1988 |
| 5 | Suzhou | EHWL | 0.06 | 0.0087 | 2.96 | ** | 1979 |
| | | ELWL | -0.02 | 0.0105 | 7.03 | ** | 1987 |
| 6 | Kunshan | EHWL | 0.31 | 0.0082 | 2.41 | ** | 1988 |
| | | ELWL | 0.67 | 0.0123 | 6.14 | ** | 1988 |
| 7 | Pingwang | EHWL | 0.17 | 0.0108 | 3.53 | ** | 1987 |
| | | ELWL | 0.36 | 0.0076 | 5.86 | ** | 1989 |
| 8 | Changshu | EHWL | 0.17 | 0.0078 | 3.48 | ** | 1985 |
| | | ELWL | 0.43 | 0.0114 | 7.65 | ** | 1988 |

Note: * and ** represent 0.05 and 0.01 significance levels, respectively



Table 3 M-K test results for monthly extreme water level in the PRNRTB

| Month | EHWL | | | ELWL | | |
|---|---|---|---|---|---|---|
| | $\beta$ | Z | Significant | $\beta$ | Z | Significant |
| Jan | 0.0115 | 5.19 | ** | 0.0096 | 5.39 | ** |
| Feb | 0.0127 | 5.66 | ** | 0.0128 | 5.91 | ** |
| Mar | 0.0144 | 5.86 | ** | 0.0129 | 5.61 | ** |
| Apr | 0.0102 | 4.68 | ** | 0.0116 | 5.64 | ** |
| May | 0.0092 | 3.92 | ** | 0.0091 | 4.87 | ** |
| Jun | 0.0134 | 3.48 | ** | 0.0095 | 4.38 | ** |
| Jul | 0.0121 | 3.23 | ** | 0.0117 | 4.90 | ** |
| Aug | 0.0159 | 4.49 | ** | 0.0135 | 4.77 | ** |
| Sep | 0.0117 | 3.24 | ** | 0.0123 | 5.06 | ** |
| Oct | 0.0065 | 2.33 | * | 0.0058 | 2.52 | ** |
| Nov | 0.0084 | 3.42 | ** | 0.0076 | 3.46 | ** |
| Dec | 0.0096 | 4.03 | ** | 0.0097 | 5.03 | ** |

Note: * and ** represent 0.05 and 0.01 significance levels, respectively.





Table 4 Contributions of precipitation and human activity on the changes in extreme water level in the PRNRTB

| Water level | Period | $H_O/H_A$ (m) | $H_{HP}$ (m) | $\Delta H_T$ (m) | $\Delta H_H$ (m) | | $\Delta H_P$ (m) | |
|---|---|---|---|---|---|---|---|---|
| | | | | | (m) | % | (m) | % |
| EHWL | 1960-1988 | 3.72 | | | | | | |
| | 1989-2000 | 4.04 | 3.86 | 0.18 | 0.04 | 20.5% | 0.14 | 79.5% |
| | 2001-2012 | 3.97 | 3.78 | 0.20 | 0.14 | 70.3% | 0.06 | 29.7% |
| | 1989-2012 | 4.00 | 3.82 | 0.19 | 0.09 | 46.7% | 0.10 | 53.3% |
| ELWL | 1960-1988 | 2.51 | | | | | | |
| | 1989-2000 | 2.77 | 2.59 | 0.18 | 0.10 | 54.8% | 0.08 | 45.2% |
| | 2001-2012 | 2.89 | 2.58 | 0.31 | 0.25 | 80.1% | 0.06 | 19.9% |
| | 1989-2012 | 2.83 | 2.59 | 0.24 | 0.17 | 70.9% | 0.07 | 29.1% |

Table 5 Changes of river networks in the PRNRTB during the past 50 years

| | 1960s | 1980s | 2010s |
|---|---|---|---|
| River length (km) | 32079.95 | 31545.31 | 28010.87 |
| Water area (km$^2$) | 1161.08 | 1073.19 | 925.92 |
| Drainage density (km/km$^2$) | 3.66 | 3.60 | 3.20 |
| Water surface ratio (%) | 13.26 | 12.26 | 10.58 |



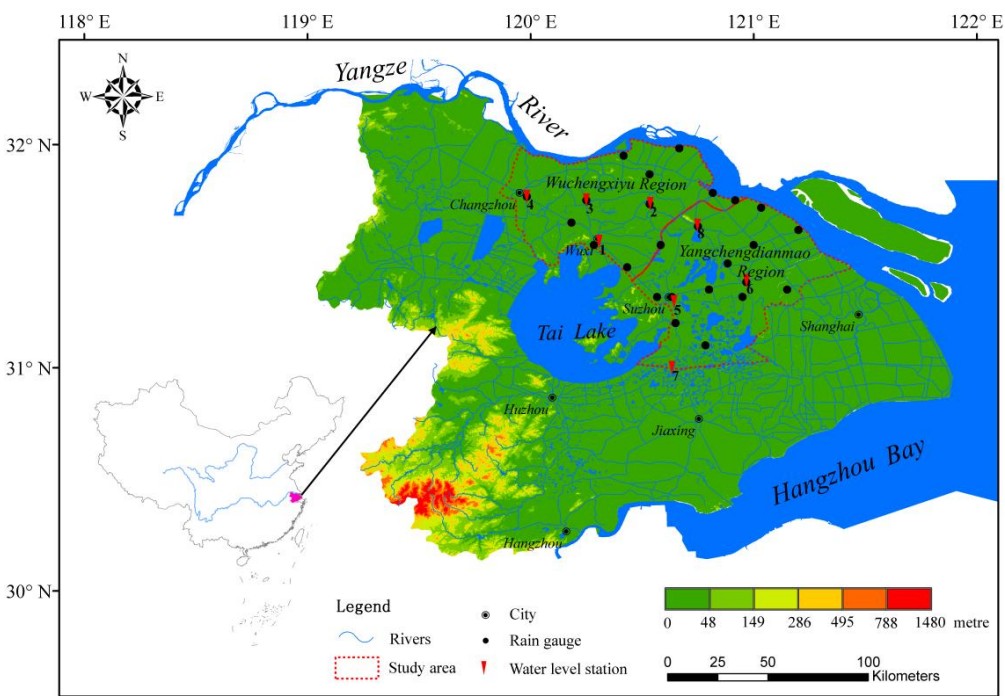

Fig. 1 Location of study area and the distribution of gauge stations

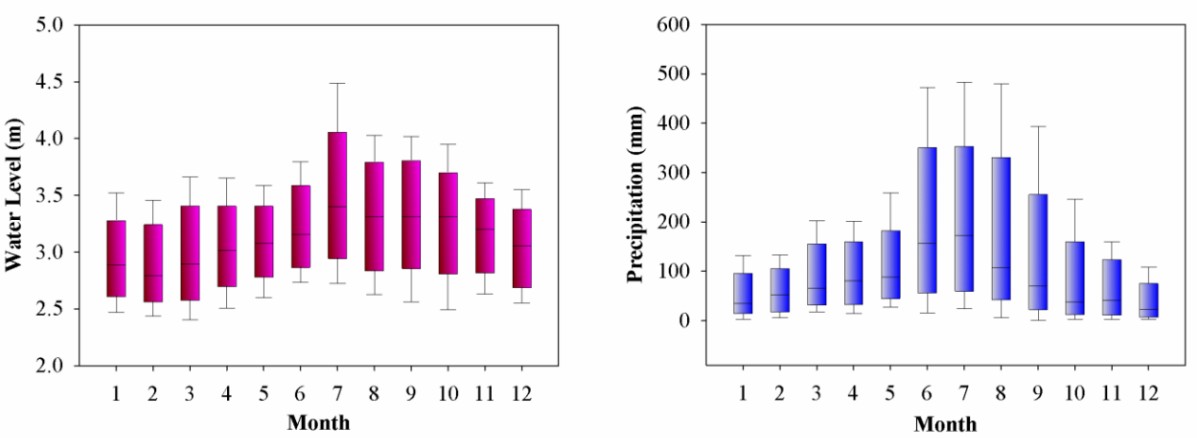

Fig. 2 Boxplots of the average water level and precipitation in the PRNRTB





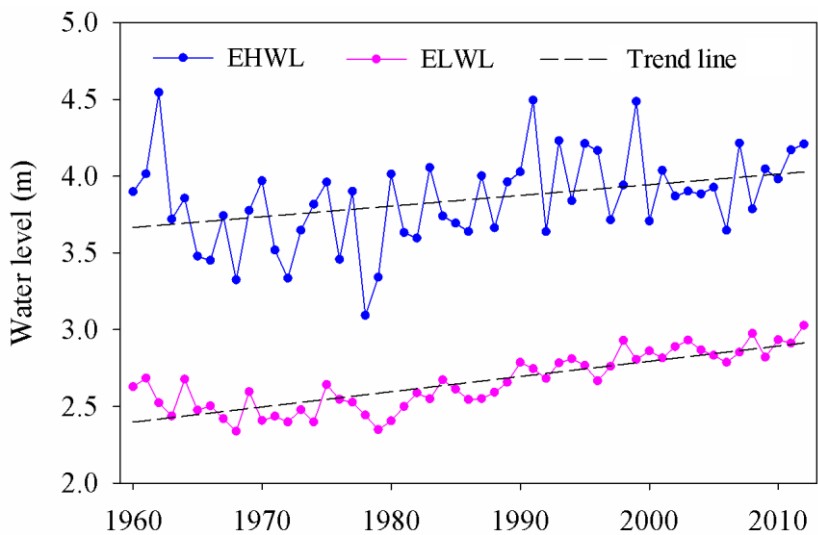

Fig. 3 Variations of annual extreme water level of the PRNRTB from 1960 to 2012

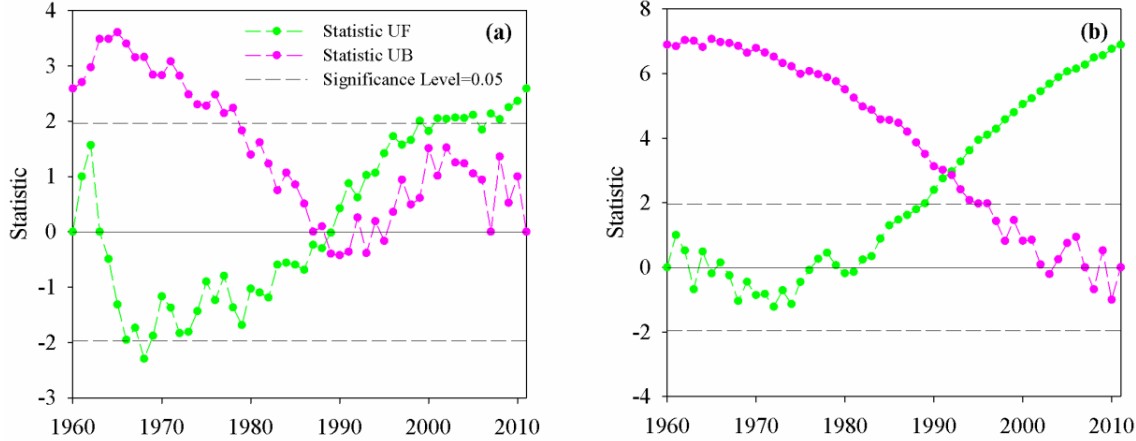

Fig. 4 M-K test for annual extreme water level in the PRNRTB from 1960 to 2012, (a) EHWL and (b) ELWL. The

green dotted line refers to the sequential statistical curve UF, and pink dot line refers to the reverse statistical curve UB




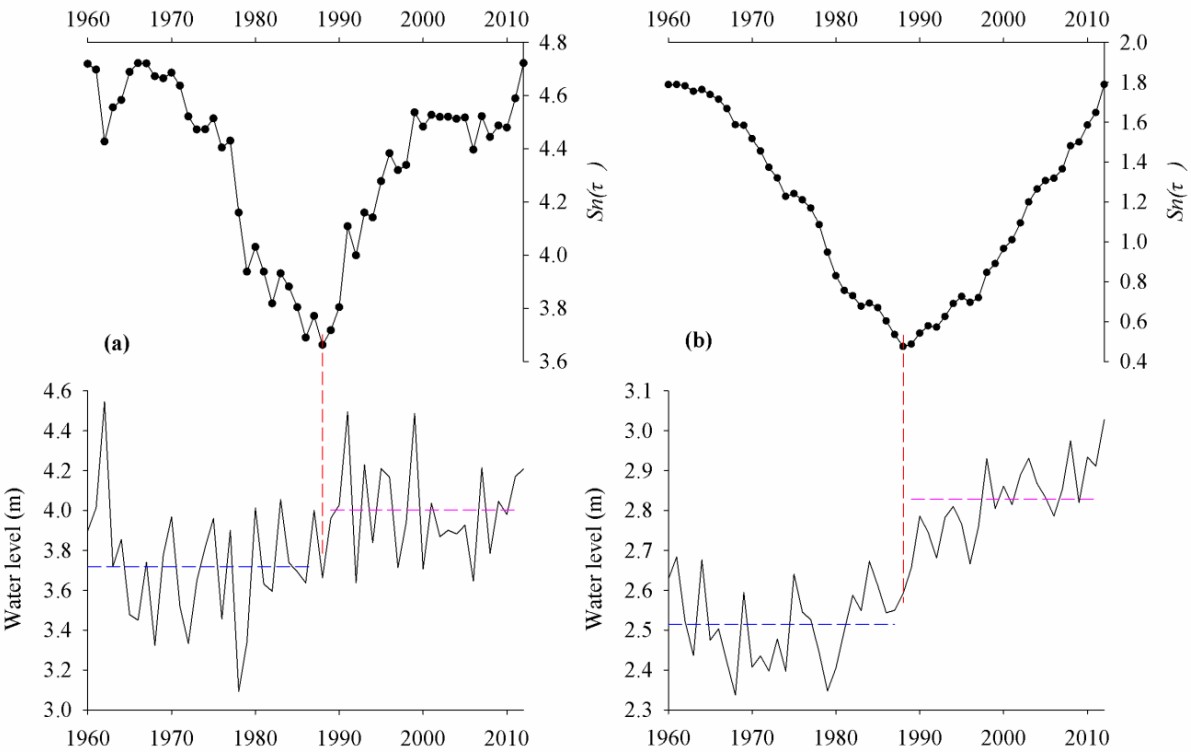

Fig. 5 Sequential cluster analysis for the extreme water level in the PRNRTB from 1960 to 2012, (a) EHWL and (b) ELWL. The red dotted line is the year of change point, blue and pink dotted lines are the average water level in different period, respectively

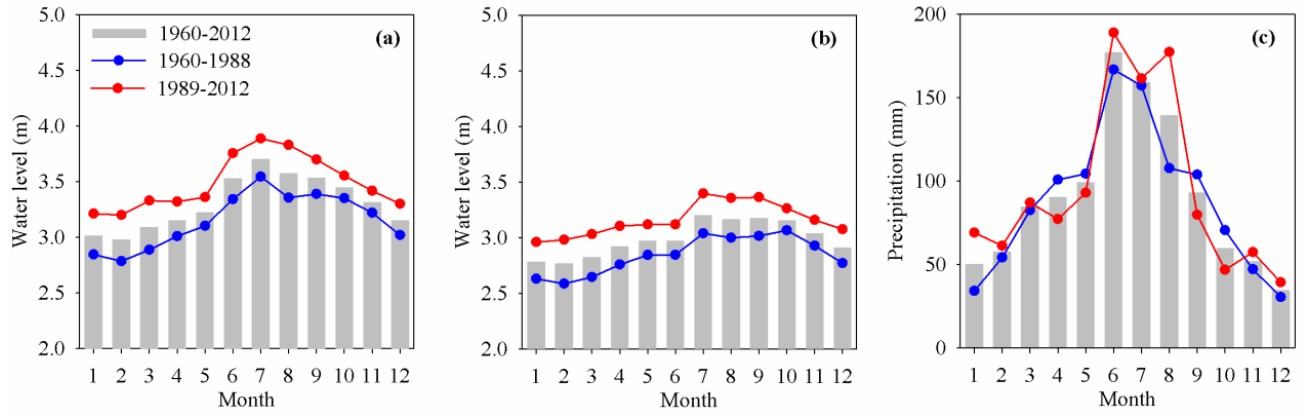

Fig. 6 Monthly extreme water level and average precipitation during three different periods in the PRNRTB, (a) EHWL, (b) ELWL and (c) Precipitation




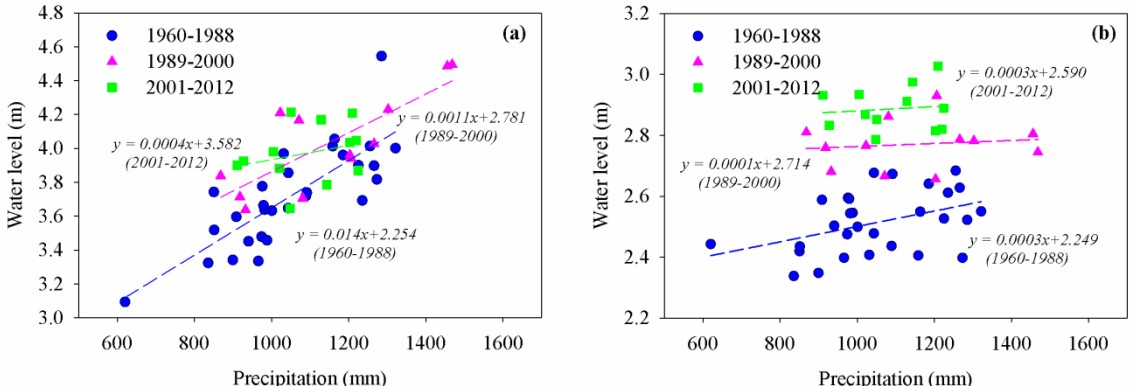

Fig. 7 Linear regression equations between extreme water level and precipitation during different periods in the PRNRTB, (a) EHWL and (b) ELWL

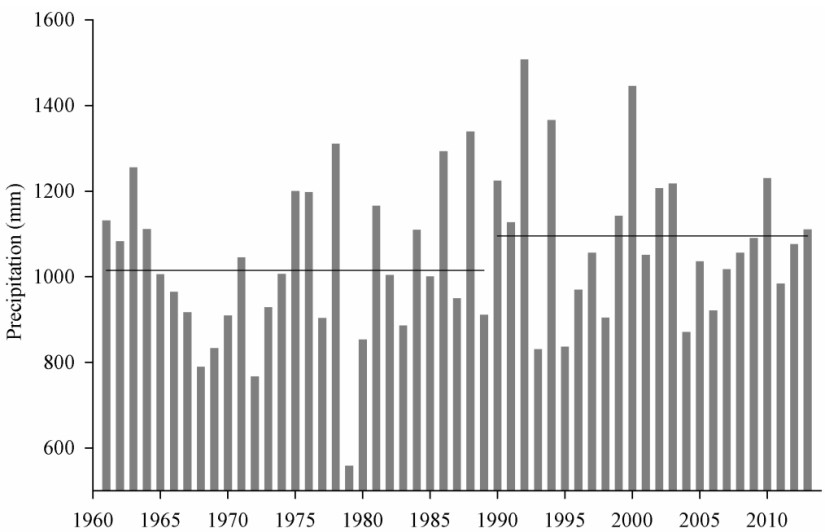

Fig. 8 Variation of annual precipitation in the PRNRTB from 1960 to 2012, the solid lines represented the average precipitation of 1960-1988 and 1989-2012.

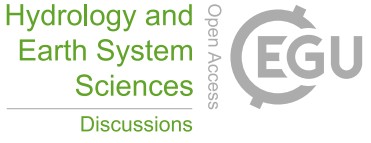

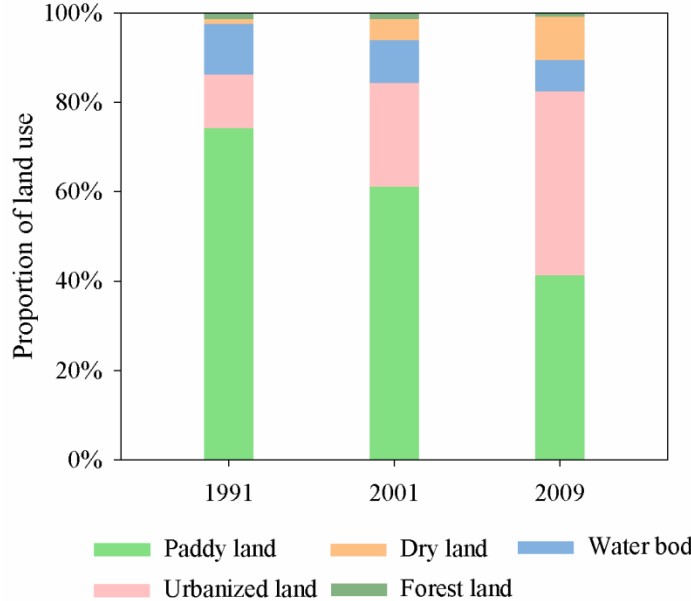

Fig. 9 Changes of proportion of land use in the PRNRTB from 1991 to 2009