# Peer review of "Changing patterns of extreme water levels in urbanizing plain river network region of Taihu Basin, China: characteristics and causes"

_Hydrology and Earth System Sciences, 2016_

## Referee Comment (RC1) · Anonymous Referee #1 · 27 May 2016

The manuscript studied the temporal trend and change point of the annual maximum and minimum water level in Taihu Basin, China. The studies used an interesting high-quality data sets which is the daily water level datasets from 1960 to 2012 at 8 stations in the basin. Despite this interesting data set, the case study considered in the manuscript is very basic, and scientific contribution looks limited. The paper also suffers from serious flaws on the methodology and some conclusions are not supported by evidence. In my opinion, the current manuscript does not meet the standard of HESS, which still requires substantial improvement.

- Page 3, line 30. The thiessen polygons method (Jones and Hulme, 1996) was used to calculate regional extreme water level series. However, these kinds of method are

usually not theoretically justified for extreme data sets (e.g. annual maximum or annual minimum), because extreme data are usually highly skewed. Thus an acceptable way to regionalizing the extreme data set is to firstly regionalizing the daily water level time series at different locations, then calculate the maximum or minimum. The reverse of this procedure is not correct.

However, owning such a high-quality data set, I suggest to consider some regional models to study at-site extreme data, rather than regionalizing to one time series, in which huge amount of information can be lost during the process of regionalization. Focusing directly the at-site data could provide a better understand on how the extreme water level changes at each site due to climate and human activities. I think it could be interesting to consider some extreme values distribution to quantify such risks. However, just reporting the existence of trend is not enough. Authors could consider some of the following references for the regional approaches on extreme events.

Leclerc, M. and T. B. M. J. Ouarda (2007). "Non-stationary regional flood frequency analysis at ungauged sites." Journal of Hydrology 343(3-4): 254-265.

Maraun, D., et al. (2011). "The influence of synoptic airflow on UK daily precipitation extremes. Part I: Observed spatio-temporal relationships." Climate Dynamics 36(1-2): 261-275.

Sun, X., et al. (2014). "A general regional frequency analysis framework for quantifying local-scale climate effects: A case study of ENSO effects on Southeast Queensland rainfall." Journal of Hydrology 512(0): 53-68.

Chen, X., et al. (2014). "Climate information based streamflow and rainfall forecasts for Huai River basin using hierarchical Bayesian modeling." Hydrology And Earth System Sciences 18(4): 1539-1548.

Steinschneider, S. and U. Lall (2015). "A hierarchical Bayesian regional model for non-stationary precipitation extremes in Northern California conditioned on tropical moisture exports." Water Resources Research 51(3): 1472-1492.

- A follow up question is that in page 9 line 12 authors mentioned heterogeneity of rainfall in Taihu Basin (Deng et al., 2014). It is not clear why regionalizing the extreme water level data in this heterogeneous region is appropriate, because regionalizing data usually requires to be in the homogeneous region.

- Page 8, line 17, the precipitation is not well defined (Monthly? Daily?) Beta and z are not defined in Table 3. Authors attempt to use rainfall as a main predictor to estimate the contribution of climate on the maximum and minimum water level. However, selecting the rainfall at different periods (e.g. summer rainfall, annual rainfall) or different events (e.g. average rainfall, n-days cumulative rainfall and daily maximum rainfall) could lead to a very different result. Thus it is not rigorous to select the predictors arbitrarily.

- Page 8, line 21-30. I disagree that the authors account for all those variation that could not be explained by rainfall as human activities. To account for the impact of human activities, authors need to conduct a quantitative investigation on each human activity, such as land use, and assess their impacts. Besides the climate and human activities, there is also a portion of variation (an error term or residual) which could not be captured by either of these aspects. Thus the methods of quantifying the impact of human activities is problematic.

Furthermore, just using rainfall to account for climate factors is also problematic. A linear regression based on rainfall (it is not clear which kinds of rainfall is used here) was used to estimate the annual minimum and maximum water level. However, Fig 7 showed that the correlation between the rainfall and water level is quite weak after 1989. Thus "rainfall" used here is certainly not a good predictor. As a consequence, just using rainfall to calculate H_HP is not appropriate. Calculating H_hp should accounts for all climate factors besides rainfall, such as evaporation.

Additional comments:

[Figure]

- Page 2, Line 23, what is km/km2?

- Page 6, lines 20. Curve of statistics UF (and also UB) have never been defined, whereas UF is sometimes used as the normalized variables (removing the mean and divide by the standard deviation). UB is similar to UF, but calculated using the reversed data. This definition needs to be clearly stated in the manuscript.

- Page 6, line 29, why obvious?

- Page 7, line 5. How to support this judgement? References are required.

---

## Referee Comment (RC2) · Anonymous Referee #2 · 31 May 2016

This paper presents an analysis of daily rainfall and water level data for the period 1960 to 2012 for the Taihu basin, in the delta region of the Yangtze river in China. The aim is to detect changes in annual minimum and maximum water levels over the study period and to see whether changes can be attributed to changes in precipitation patterns and/or to anthropogenic impacts. While the dataset is rich in that it includes data for 24 rain gauges and 8 water level gauges, most of the analyses seem to be based on single water level and rainfall time series, representing either a single gauge or an average over all gauges (this is not clear from the text). Basic statistical tests are performed for trend analysis and detection of change points in the time series of annual maxima and minima. Then, an attempt is made to attribute detected changes

to precipitation and anthropogenic influence. The latter is based on small datasets (the original dataset is split into two) , which provides a weak and, in my view insufficient, basis to support conclusions drawn from this analysis.

The paper would greatly benefit from more extensive analysis of the available datasets, including an explicit analysis of data from the individual gauges and looking more deeply into relationships between rainfall and water levels across the basin. One of the key aspects of this study is analysis of anthropogenic impacts on the hydrological system, which are associated with rapid urbanisation over the past decades. This relationship is studied in a rather indirect way (based on the annual data series) and has a spatial aspect that is not touched upon in the paper. With the available data, there seems to be room for analysis of spatially varied urbanisation impacts across the basin (since it is likely that urbanisation is non-homogeneously distributed across the basin, one would expect different impacts on water levels at different gauges over time).

Comments per section: Introduction: - In their discussion of references, the authors refer to water level as one of the relevant hydrological parameters. The reason why they do so, is because they will be analysing water levels, not flows, in their study. This is however not explicitly mentioned, which makes the description in the first 2 paragraphs somewhat confusing. I would suggest to first present relevant literature, then explain specifics of water level instead of flow as a relevant parameter for delta regions, where storage (water level variation) dominates over flow. - P2, line 23: what do the authors mean by "criss-cross river network with a density of 3.2 km/km2", is this a river network density? In general, this paragraph (lines 23-35) needs to be restructured in a more logical way Study area and data source: - P3, lines 20-21: please report period for which the percentages are reported - P3, line 30: it is stated that Thiessen polygons were used to calculate regional extreme water level series. First, this does not seem to be an appropriate method for interpolation of water levels and second, it is not clear why this interpolation was done, since later analyses seem to be based on single data series? - P3, lines 27 and 32: please report type of gauges for rainfall and water levels,

data resolution and data gaps, if any. - P4, lin1: please specify "some other data" – what land use data, river network data and how these were used in the analysis Methodology: - In section 3.2, the dataset is split into 2 separate periods to study water level changes. The resulting sub-datasets are very small in size, thus insufficient to support statistical analysis. Also, the assumed relationships between precipitation and water level are far too simplistic to draw valid conclusions from the results. Results and discussion: - At several points in the discussion, authors draw conclusions on the impact of climate change and human activity on water levels which are not well justified by the results. A more critical analysis and discussion of results is required (for instance on page 6, lines 25-26; page 7, lines 10-11 and lines 26-27) - Section 4.2: as mentioned earlier, the number of data points seems to be too small to draw these conclusions - Section 4.3: lines 27: please check numbers for urbanisation, they do not match and seem to be rather low compared to the size of the basin (7929 km2)

---

## Referee Comment (RC3) · Anonymous Referee #3 · 23 Jun 2016

The paper attempts to investigate the influence of climate change and human activity on changes of water levels in the plain river network region of Taihu Basin, China. Daily water level data from 8 monitoring stations and for the period 1962-2012 have been used in the analysis. Two time series representing annual maximum water levels and annual minimum water levels have been derived from the daily data for the 8 stations. One single regionalisation series for each of the two variables has then been calculated by averaging the time series from the 8 monitoring stations using the Thiessen polygons method. The resulting two regionalisation series have been analysed using basic statistical methods used in testing the significance of trend changes in a time series. I believe that averaging of the 8 time series in one single regional

series represents a major flaw in the methodology of the paper. Some of the significant variations in each of the 8 time series could be smoothed out by this averaging and hence false results are obtained from the analysis. Therefore I suggest that the authors redo the analysis on each of the 8 time series separately and use the results of this analysis in answering the main research questions of the paper similar to the approach used by Murphey et al. (2013).

Murphy, C., Harrigan, S., Hall, J., and Wilby, R.L. (2013). Climate-driven trends in mean and high flows from a network of reference stations in Ireland. Hydrological Sciences Journal – Journal des Sciences Hydrologiques, 58 (4) 2013 755. http://dx.doi.org/10.1080/02626667.2013.782407.

Some specific comments:

Mann-Kendall statistic Z and two statistics UF and UB used in Figure 4 need to be described in Section 3.1.1.

In section 4.2 the authors need to justify why is the activity period (between 1989-2000) divided into two shorter periods.

More international references must be cited in the paper.

---

## Author Comment (AC1) · 4 Aug 2016

The manuscript studied the temporal trend and change point of the annual maximum and minimum water level in Taihu Basin, China. The studies used an interesting high quality data sets which is the daily water level datasets from 1960 to 2012 at 8 stations in the basin. Despite this interesting data set, the case study considered in the manuscript is very basic, and scientific contribution looks limited. The paper also suffers from serious flaws on the methodology and some conclusions are not supported by evidence. In my opinion, the current manuscript does not meet the standard of HESS, which still requires substantial improvement.

Author answer: We really appreciate this reviewer's very valuable and constructive

advice on our manuscript. It will greatly help us to improve the quality of our manuscript. As the reviewer said, we conducted a basic research in this manuscript. However, as far as our knowledge, such study on long-term changes in extreme water level in this region is very limited. In recent decades, due to the rapid urbanization, the hydrological regimes have been changed heavily in this region. Therefore, we take comprehensive researches on the changes in water level at first. Certainly, as the reviewers expected, some further work on rainstorm modeling and flood risk will go on in our further work.

As these general comments are reflected in the following specific comments, we will address them as follows:

Comment 1: - Page 3, line 30. The thiessen polygons method (Jones and Hulme, 1996) was used to calculate regional extreme water level series. However, these kinds of method are usually not theoretically justified for extreme data sets (e.g. annual maximum or annual minimum), because extreme data are usually highly skewed. Thus an acceptable way to regionalizing the extreme data set is to firstly regionalizing the daily water level time series at different locations, then calculate the maximum or minimum. The reverse of this procedure is not correct.

Author answer: Thanks for this comment. As the Reviewer suggested, it is not appropriate that we select the thiessen polygons method to calculate the regional extreme water level. According to the comment, we will reanalyze the characteristic of extreme water level through at-site data in our revised manuscript.

Comment 2: However, owning such a high-quality data set, I suggest to consider some regional models to study at-site extreme data, rather than regionalizing to one time series, in which huge amount of information can be lost during the process of regionalization. Focusing directly the at-site data could provide a better understand on how the extreme water level changes at each site due to climate and human activities. I think it could be interesting to consider some extreme values distribution to quantify such risks. However, just reporting the existence of trend is not enough. Authors could consider some of the following references for the regional approaches on extreme events. Leclerc, M. and T. B. M. J. Ouarda (2007). "Non-stationary regional flood frequency analysis at ungauged sites." Journal of Hydrology 343(3-4): 254-265. Maraun, D., et al. (2011). "The influence of synoptic airflow on UK daily precipitation extremes. Part I: Observed spatio-temporal relationships." Climate Dynamics 36(1-2): 261-275. Sun, X., et al. (2014). "A general regional frequency analysis framework for quantifying local-scale climate effects: A case study of ENSO effects on Southeast Queensland rainfall." Journal of Hydrology 512(0): 53-68. Chen, X., et al. (2014). "Climate information based streamflow and rainfall forecasts for Huai River basin using hierarchical Bayesian modeling." Steinschneider, S. and U. Lall (2015). "A hierarchical Bayesian regional model for on stationary precipitation extremes in Northern California conditioned on tropical moisture exports." Water Resources Research 51(3): 1472-1492.

Author answer: According to the Reviewer's suggestion, we will reanalyze the characteristic of water level through at-site data. What's more, we also have read the recommended references carefully. We also think it's very interesting and meaningful to quantify flood risk in this region combining regional model and extreme values theory. Such study can help us better understand the characteristic of extreme events. I think this valuable advice points out the direction for our further study.

Comment 3: - A follow up question is that in page 9 line 12 authors mentioned heterogeneity of rainfall in Taihu Basin (Deng et al., 2014åřŚäžʀè£ŹçŕĞæŰĞçŇő). It is not clear why regionalizing the extreme water level data in this heterogeneous region is appropriate, because regionalizing data usually requires to be in the homogeneous region.

Author answer: Thanks for this comment. We mentioned the heterogeneity of rainfall in this reference refers to uneven distribution at the time scale. As we all know, the Taihu Basin is controlled by the East Asia Monsoon and the precipitation is mainly concentrated in the wet season (May to September). According to Deng et al., (2014), the concentration of precipitation in wet season has an uptrend, which may contribute

to water level increasing. What's more, we will reanalyze the characteristic of extreme water level through at-site time series in our revised manuscript.

Comment 4: - Page 8, line 17, the precipitation is not well defined (Monthly? Daily?) Beta and z are not defined in Table 3. Authors attempt to use rainfall as a main predictor to estimate the contribution of climate on the maximum and minimum water level. However, selecting the rainfall at different periods (e.g. summer rainfall, annual rainfall) or different events (e.g. average rainfall, n-days cumulative rainfall and daily maximum rainfall) could lead to a very different result. Thus it is not rigorous to select the predictors arbitrarily.

Author answer: Thanks for this valuable advice. Actually, we selected the annual precipitation as predictor to estimate the maximum and minimum water level in our original manuscript. According to the the statistical results, we found that the maximum and minimum water level occurred during the wet and dry season, respectively. Considering the Reviewer's suggestion, we will use the climate factors of wet season to estimate the maximum water level and use the climate factors of dry season to estimate the minimum water level. Besides precipitation, the evaporation also will be considered as a climate factor to estimate water level. Details can be referred to Comment 6. In addition, we will clarify these definitions (Beta and Z) in our revised manuscript according to the Reviewer's suggestion,

Comment 5: - Page 8, line 21-30. I disagree that the authors account for all those variation that could not be explained by rainfall as human activities. To account for the impact of human activities, authors need to conduct a quantitative investigation on each human activity, such as land use, and assess their impacts. Besides the climate and human activities, there is also a portion of variation (an error term or residual) which could not be captured by either of these aspects. Thus the methods of quantifying the impact of human activities is problematic.

Author answer: Thanks for this comment. In this study, we use this method to investi-

gate whether climate change or anthropogenic factor should be responsible for hydrological variation at different periods across the basin. Actually, the Reviewer's advice is very constructive for understanding the hydrological variation caused by human activity. As we all know, this region has experienced many kinds of human activities due to urbanization during the recent decades, such as land use changes, river network reduction, and hydraulic engineering constructions. It is very interesting and meaningful to investigate hydrological variation from each human activity on in this region. Because it's very difficult to investigation their separate impacts on water level alteration across the entire region due to the limited of human activity data. But I think it's feasible to carry out such research in a small basin through hydrologic model and the Reviewer's advice points out the direction for our further study. However, urbanization area is not homogeneously distributed in this region. We will try to explore what relationship it is between urbanization distribution and the alteration of water level in our revised manuscript.

Comment 6: Furthermore, just using rainfall to account for climate factors is also problematic. A linear regression based on rainfall (it is not clear which kinds of rainfall is used here) was used to estimate the annual minimum and maximum water level. However, Fig 7 showed that the correlation between the rainfall and water level is quite weak after 1989. Thus "rainfall" used here is certainly not a good predictor. As a consequence, just using rainfall to calculate H_HP is not appropriate. Calculating H_hp should accounts for all climate factors besides rainfall, such as evaporation.

Author answer: Thanks for this valuable suggestion. According to the Reviewer's advice, it is not appropriate that we just used annual precipitation to estimate extreme water level. So we will select the precipitation and evaporation to rebuild the relationship between extreme water level and climate factors during the baseline period in the revised manuscript. Then, we will reanalyze water level variability attribution through the relationship equation. In original manuscript (Fig 7), the correlation coefficient between extreme water level and annual precipitation is high during the baseline period

(1960-1988), so we only used the equation in the baseline period to calculate HHP in activity period. Considering the Reviewer's suggestion, the whole time series will be divided according to decade scale (i.e. 1960-1969, 1970-1979, 1980-1989, 1990-1999, 2000-2012). Meanwhile, Fig 7 will be modified and some relevant paragraphs also will be rewritten in our revised manuscript.

Additional comments:

Comment 7: - Page 2, Line 23, what is km/km2?

Author answer: It is the unit of river density. River density is the total length (km) of river networks in unit basin area (km2), which was firstly introduced by Horton (1945). We will rewrite this sentence and add related notes in our revised manuscript.

Comment 8: - Page 6, lines 20. Curve of statistics UF (and also UB) have never been defined, whereas UF is sometimes used as the normalized variables (removing the mean and divide by the standard deviation). UB is similar to UF, but calculated using the reversed data. This definition needs to be clearly stated in the manuscript.

Author answer: We agree. We will add the definition of statistics UF and UB in our revised paragraph according to the Reviewer's advice. Comment 9: - Page 6, line 29, why obvious?

Author answer: According to the Yue et al., (2002), the existence of serial correlation may increases the probability that the Mann-Kendall test will detect a significant trend. In this study, the lag-1 R of extreme high water level (EHWL) is lower than that of extreme low water level (ELWL). It is because that EHWL is a strong random sample, the occurrences of which are usually accompanied by heavy precipitation. The ELWL occurs in the dry season and its value range is smaller than that of EHWL, which lead to high serial correlation. We will revise this sentence in our revised paragraph to enhance readability.

Comment 10: - Page 7, line 5. How to support this judgement? References are required.

Author answer: We will add relevant references in this sentence when we revise it.

---

## Author Comment (AC2) · 4 Aug 2016

This paper presents an analysis of daily rainfall and water level data for the period 1960 to 2012 for the Taihu basin, in the delta region of the Yangtze river in China. The aim is to detect changes in annual minimum and maximum water levels over the study period and to see whether changes can be attributed to changes in precipitation patterns and/or to anthropogenic impacts. While the dataset is rich in that it includes data for 24 rain gauges and 8 water level gauges, most of the analyses seem to be based on single water level and rainfall time series, representing either a single gauge or an average over all gauges (this is not clear from the text). Basic statistical tests are performed for trend analysis and detection of change points in the time series of

annual maxima and minima. Then, an attempt is made to attribute detected changes to precipitation and anthropogenic influence. The latter is based on small datasets (the original dataset is split into two), which provides a weak and, in my view insufficient, basis to support conclusions drawn from this analysis.

Author answer: We really appreciate this Reviewer's valuable and constructive advice on our manuscript. It will greatly help us to improve the quality of our manuscript. In original manuscript, the analysis is mainly based on the regional average water level. After considering the Reviewer's suggestion, we found it was not appropriate to study extreme water level like that. We will reanalyze extreme water level through single water level series in our revised manuscript. What's more, in Section 4.2, we will divide the whole time series according to decade scale (i.e. 1960-1969, 1970-1979, 1980-1989, 1990-1999, 2000-2012) and relevant paragraphs also will be rewritten.

The paper would greatly benefit from more extensive analysis of the available datasets, including an explicit analysis of data from the individual gauges and looking more deeply into relationships between rainfall and water levels across the basin. One of the key aspects of this study is analysis of anthropogenic impacts on the hydrological system, which are associated with rapid urbanisation over the past decades. This relationship is studied in a rather indirect way (based on the annual data series) and has a spatial aspect that is not touched upon in the paper. With the available data, there seems to be room for analysis of spatially varied urbanisation impacts across the basin (since it is likely that urbanisation is no-homogeneously distributed across the basin, one would expect different impacts on water levels at different gauges over time).

Author answer: Thanks for the constructive comments. As we all know, urbanization area is not homogeneously distributed across this region. Considering the Reviewer's advice, we will try to explore what relationship it is between urbanization distribution and the alteration of water level in our revised manuscript.

Comments per section:

1. Introduction: - In their discussion of references, the authors refer to water level as one of the relevant hydrological parameters. The reason why they do so, is because they will be analysing water levels, not flows, in their study. This is however not explicitly mentioned, which makes the description in the first 2 paragraphs somewhat confusing. I would suggest to first present relevant literature, then explain specifics of water level instead of flow as a relevant parameter for delta regions, where storage (water level variation) dominates over flow.

Author answer: Thanks, this valuable comment will make the introduction more logical. We will reconstruct this section according to the Reviewer's suggestion.

2. - P2, line 23: what do the authors mean by "criss-cross river network with a density of 3.2 km/km2", is this a river network density? In general, this paragraph (lines 23-35) needs to be restructured in a more logical way.

Author answer: Thanks for this comment. It is the unit of river density. River density is the total length (km) of river networks in unit basin area (km2), which was firstly introduced by Horton (1945). We also will rewrite this paragraph in a more logical way when we revise it.

3. Study area and data source: - P3, lines 20-21: please report period for which the percentages are reported.

Author answer: We will rewrite this sentence in revised manuscript according to the Reviewer's suggestion.

4. - P3, line 30: it is stated that Thiessen polygons were used to calculate regional extreme water level series. First, this does not seem to be an appropriate method for interpolation of water levels and second, it is not clear why this interpolation was done, since later analyses seem to be based on single data series?

Author answer: Thanks for this valuable advice. In original manuscript, the analysis is mainly based on the regional extreme water level series, and some trend analysis was

also detected for single series. After considering the Reviewer's suggestion, we found it was not appropriate to study extreme water level like that. We will reanalyze extreme water level through single water level series in our revised manuscript. Moreover, some relevant paragraphs also will be rewritten.

5. - P3, lines 27 and 32: please report type of gauges for rainfall and water levels, data resolution and data gaps, if any.

Author answer: In this study, the water level and rainfall are both daily scale data. Both of them are with no data gap. Water level is collected from eight national hydrologic station and rainfall data is collected from national precipitation station. In addition, we also will add some sentences to make data sources clear.

6. - P4, lin1: please specify "some other data" – what land use data, river network data and how these were used in the analysis Methodology.

Author answer: Thanks for this comment. In this study, land use and river network data are mainly used to reflect changes in human activities. Due to the rapid urbanization, landscape and river system have been changed dramatically in this region. Land use data and the area of each type were extracted from three periods of TM images in 1991, 2001 and 2010 provided by the US Geological Survey. River networks were extracted using three periods of topographic maps (1:50000) in the 1960s, 1980s and 2010s. These human activities have an important effect on hydrological factors, such as water level. Therefore, we conduct a comprehensive analysis on these data in discussion section. What's more, some specifications on these data will be added when we revise it.

7. - In section 3.2, the dataset is split into 2 separate periods to study water level changes. The resulting sub-datasets are very small in size, thus insufficient to support statistical analysis. Also, the assumed relationships between precipitation and water level are far too simplistic to draw valid conclusions from the results.

Author answer: Thanks for this valuable suggestion. Considering the Reviewer's suggestion, we will divide the entire period according to decade scale (i.e. 1960-1969, 1970-1979, 1980-1989, 1990-1999, 2000-2012). According to the Reviewer's advice, it is not appropriate that we just used annual precipitation to estimate extreme water level. So we will select the precipitation and evaporation to build the relationship between extreme water level and climate factors, and the relationship equations also will be built according to dry and wet seasons respectively, which means that the climate factors of wet season are used to estimate maximum water level and the climate factors of dry season are used to estimate the minimum water level. In addition, relevant paragraphs also will be rewritten in our revised manuscript.

8. Results and discussion: - At several points in the discussion, authors draw conclusions on the impact of climate change and human activity on water levels which are not well justified by the results.

Author answer: Thanks for the comment. In Section 4.3, we have made a comprehensive discussion about climate change and human activity on water level changes in this region. According to the Reviewer's suggestion, we will reconsider some expression of conclusion and reconstruct the paragraphs according to reanalysis results. Meanwhile, some necessary references also will be cited to make the manuscript more logically and clearly.

9. A more critical analysis and discussion of results is required (for instance on page 6, lines 25-26; page 7, lines 10-11 and lines 26-27)

Author answer: Thanks for the comment. We will rewrite these sentences and add more critical analysis in our revised manuscript.

10. - Section 4.2: as mentioned earlier, the number of data points seems to be too small to draw these conclusions.

Author answer: Thanks for this comment. Considering the Reviewer's suggestion,

we will divide the period according to decade scale. Some details can be referred to Comment 7.

11. - Section 4.3: lines 27: please check numbers for urbanisation, they do not match and seem to be rather low compared to the size of the basin (7929 km2).

Author answer: Thanks for this comment. We have made a mistake about it. The urban land area should be 2481 km2. We have corrected it in our revised manuscript.

———————————————————

---

## Author Comment (AC3) · 4 Aug 2016

Comment 1: The paper attempts to investigate the influence of climate change and human activity on changes of water levels in the plain river network region of Taihu Basin, China. Daily water level data from 8 monitoring stations and for the period 1962-2012 have been used in the analysis. Two time series representing annual maximum water levels and annual minimum water levels have been derived from the daily data for the 8 stations. One single regionalisation series for each of the two variables has then been calculated by averaging the time series from the 8 monitoring stations using the Thiessen polygons method. The resulting two regionalisation series have been analysed using basic statistical methods used in testing the significance of trend changes

in a time series. I believe that averaging of the 8 time series in one single regional series represents a major flaw in the methodology of the paper. Some of the significant variations in each of the 8 time series could be smoothed out by this averaging and hence false results are obtained from the analysis.

Author answer: We really appreciate this Reviewer's valuable and constructive advice on our manuscript. It will greatly help us to improve the quality of our manuscript. In original manuscript, the analysis is mainly based on the regional water level series. After considering the Reviewer's suggestion, we found it was not appropriate to study extreme water level like that. We will reanalyze extreme water level based on single water level series in our revised manuscript.

Comment 2: Therefore I suggest that the authors redo the analysis on each of the 8 time series separately and use the results of this analysis in answering the main research questions of the paper similar to the approach used by Murphey et al. (2013). Murphy, C., Harrigan, S., Hall, J., and Wilby, R.L. (2013). Climate-driven trends in mean and high flows from a network of reference stations in Ireland. Hydrological Sciences Journal, 58 (4) 2013 755. http://dx.doi.org/10.1080/02626667.2013.782407.

Author answer: Thanks for this valuable suggestion. We have read this paper carefully, which is very useful for revising our manuscript. According to the Reviewer's advice, we will redo the analysis on each of the 8 hydrological stations separately in our revised manuscript. What's more, some paragraphs also will be rewritten according to the new analysis results.

Comment 3: Mann-Kendall statistic Z and two statistics UF and UB used in Figure 4 need to be described in Section 3.1.1.

Author answer: UF is sometimes used as the normalized variables (removing the mean and divide by the standard deviation). UB is similar to UF, but calculated using the reversed data. This definition will be clearly described in the revised manuscript.

Comment 4: In section 4.2 the authors need to justify why is the activity period (between 1989-2000) divided into two shorter periods.

Author answer: Thanks for the comment. In original manuscript, we want to analyze the changes in contribution of human activity to the extreme water level in different periods, so the activity period is divided into two shorter periods. However, considering the Reviewer's advice, we will divide the entire period according to decade scale (i.e. 1960-1969, 1970-1979, 1980-1989, 1990-1999, 2000-2012) and reconstruct the relationship equation between climate factors and water level. In addition, these paragraphs also will be rewritten in our revised manuscript.

Comment 5: More international references must be cited in the paper.

Author answer: Thanks for the comment. We will cite more international references to make the paper logical and clear when we revise it.